# Review: Ischemia Reperfusion Injury—A Translational Perspective in Organ Transplantation

**DOI:** 10.3390/ijms21228549

**Published:** 2020-11-13

**Authors:** André Renaldo Fernández, Rodrigo Sánchez-Tarjuelo, Paolo Cravedi, Jordi Ochando, Marcos López-Hoyos

**Affiliations:** 1Immunology, Universitary Hospital Marqués de Valdecilla- Research Institute IDIVAL Santander, 390008 Santander, Spain; arenaldo137@gmail.com; 2Department of Oncological Sciences, Icahn School of Medicine at Mount Sinai, New York, NY 10029, USA; rodrigo.sanchez@mssm.edu (R.S.-T.); jochando@isciii.es (J.O.); 3Immunología de Trasplantes, Centro Nacional de Microbiología, Instituto de Salud Carlos III, 28220 Majadahonda (Madrid), Spain; 4Department of Medicine, Division of Nephrology, Icahn School of Medicine at Mount Sinai, New York, NY 10029, USA; paolo.cravedi@mssm.edu; 5Red de Investigación Renal (REDINREN), 28040 Madrid, Spain

**Keywords:** ischemia reperfusion injury, cell metabolism, innate immunity, hypoxia, cell death, RNA interference

## Abstract

Thanks to the development of new, more potent and selective immunosuppressive drugs together with advances in surgical techniques, organ transplantation has emerged from an experimental surgery over fifty years ago to being the treatment of choice for many end-stage organ diseases, with over 139,000 organ transplants performed worldwide in 2019. Inherent to the transplantation procedure is the fact that the donor organ is subjected to blood flow cessation and ischemia during harvesting, which is followed by preservation and reperfusion of the organ once transplanted into the recipient. Consequently, ischemia/reperfusion induces a significant injury to the graft with activation of the immune response in the recipient and deleterious effect on the graft. The purpose of this review is to discuss and shed new light on the pathways involved in ischemia/reperfusion injury (IRI) that act at different stages during the donation process, surgery, and immediate post-transplant period. Here, we present strategies that combine various treatments targeted at different mechanistic pathways during several time points to prevent graft loss secondary to the inflammation caused by IRI.

## 1. Introduction

Organ shortage remains a major unresolved problem in organ transplantation. There is an increasing demand for organ transplantation worldwide, but there are not enough organs to meet the increasing demand. To increase the pool of organs available for transplantation the use of extended criteria to include donation after circulatory death (DCD) has been recently adopted [1,2]. Although greatly needed, these extended-criteria organs are known to suffer from more ischemic damage, which is in turn associated with suboptimal results [3,4]. In kidney transplantation, ischemia/reperfusion injury (IRI) is known to underlie the clinical entity of delayed graft function (DGF) [5]. The challenge is to be able to use those organs without compromising early and long-term success. To accomplish this, there is an urgent need to better understand the mechanistic insights into the molecular and cellular events triggered by IRI.

IRI is a multifactorial inflammatory condition with underlying factors that include hypoxia, metabolic stress, leukocyte extravasation, cellular death pathways, and activation of the immune response. In this review, we describe the main pathways involved in IRI and enumerate various mechanisms of action of previously known IRI mediators. Of note, these underlying pathways are common to most tissues, although they might have different effects depending on the organ. In addition, we summarize potential therapeutic approaches aimed at reducing the detrimental effects of IRI in both preclinical experimental studies and in human clinical trials.

## 2. Metabolic Stress

The initiating insult during ischemia injury is hypoxia (Figure 1). Low oxygen levels deplete ATP in mitochondria and switch to anaerobic cellular metabolism. Decreased ATP impairs normal functioning of the Na/K pump, sequestering Na inside the cell, and it impairs calcium excretion. The intracellular calcium overload generates reactive oxygen species (ROS) and the activation of NADPH oxidase. To prevent the detrimental effects of these molecules, Yücel et al. blocked the function of NADPH oxidase using apocynin, a natural NADPH oxidase inhibitor found in plants [6]. The study demonstrated that apocynin strengthens the antioxidant defensive system, increases glutathione production, and limits the cellular stress derived from ischemia in an experimental animal model.

ROS production was previously thought to be due to non-specific dysregulation at the electron transport chain level [7]. However, Chouchani et al. [8] showed that during ischemia, a pool of succinate accumulates and drives the reverse action of the electron transport chain complex I, generating superoxides. This mechanism can also be inhibited by rotenone, which prevents ROS production, thus interfering with the pro-inflammatory response that drives IL-1B production in activated macrophages [9]. Although direct succinate removal in the donor organ has not been studied, a more proximal intervention that minimizes the harmful effects of IRI has provided promising results. In a randomized clinical trial of 40 liver recipients, α-lipoic acid (ALA) was given before liver reperfusion; then, qPCR was used to evaluate the activation of a panel of protective genes involved in hypoxia and ROS production after reperfusion [10]. The study demonstrated that ALA inactivates ROS, regenerates endogenous antioxidants, such as glutathione and vitamin E, chelates metals, and repairs tissue damage due to oxidative stress. Moreover, it regulates protective signals during IRI, such as the hypoxia inducible factor 1 alpha (HIF-1a), which has been shown to protect from IRI through transcriptional activation of the Heme Oxygenase-1 (HO-1) [11,12]. ROS and the calcium overload produced during the initial metabolic stress results in the activation of injury pathways that lead to cell death associated with IRI. Mechanistically, IRI induces an accumulation of lactate, which decreases intracellular pH and causes conformational changes in proteins. In addition, ROS mediates lipid peroxidation and destruction of the cell membrane bilayer, while the calcium increase inside mitochondria leads to membrane instability and the release of cytochrome C. All these processes ultimately induce cell death during IRI [13].

## 3. Cell Death

IRI-induced cellular death can be regulated (apoptosis) or not regulated (necrosis), although a new pathway named necroptosis, a form of regulated necrosis, has recently been elucidated. Another newly discovered cell-death pathway is ferroptosis, which is an iron-dependent apoptosis that results in lipid peroxidation [14,15]. These molecular pathways apply not just to IRI in solid organ transplantion but to a wide array of human diseases. In animal models of IRI, several molecules involved in apoptosis have been targeted (Figure 1). Caspases mediate apoptosis through both intrinsic and extrinsic pathways. The extrinsic pathway depends on extracellular molecules such as the pro-inflammatory cytokine TNFα. This binds to death receptors in the cell membrane, which triggers a conformational change in the death-induced signaling complex (DISC) that converts procaspases to caspases. One of these extrinsic pathway caspases, caspase 8, has been investigated in a murine model of IRI. Tubular epithelial cells in the kidney are the most sensitive to hypoxia and the first to undergo apoptosis during IRI. Using siRNA directed to caspase 8, Du et al. [16] were able to make tubular epithelial cells resistant to apoptosis. Other caspases have also been targeted, such as caspase 3, which is also known as the “executioner caspase” [17,18]. Although caspase targets have not yet been evaluated in humans, studies in pigs have showed similar results in preventing apoptosis [19]. A similar siRNA approach targets the “guardian of the genome” p53, which responds to hypoxia and oxidative stress by acting as a transcription factor for genes that mediate apoptosis. While p53 exerts a protective role during tumor development, it becomes harmful in the setting of IRI, and several studies in animal models of IRI have used siRNA against p53 to inhibit the apoptosis of proximal tubule epithelial cells with encouraging results [20,21]. Cellular death is of particular interest in the setting of pancreatic transplantation, where islet cells that restore beta cell mass emerged as a promising treatment [22]. After isolation and purification, their viability is compromised especially in the immediate post-transplant period, because these cells are very sensitive to necrosis. Using human cells in vitro, Bruni et al. [22] demonstrated the ability of the cell death inhibitor ferrostatin-1 to inhibit ferroptosis and recover the compromised islet cell function. Although future in vivo and human studies are warranted, this study represents a step forward toward consolidating islet transplantation as a treatment of choice in diabetes. Ferrostatins have also been used in experimental IRI models and Linkermann et al. [13] demonstrated that kidney tubular epithelial cells do not undergo necroptosis after the loss of caspase 8. Surprisingly, the study indicated that cell death occurs mainly through ferroptosis, which mediates synchronized tubular necrosis and contributes to immune cell extravasation under ischemic conditions. Consequently, the use of ferrostatins was able to exert strong protection even in conditions of severe ischemia. Overall, the data suggest that preventing the injurious effects of increased cell death during IRI represents a promising approach to increase the quality of organs available for transplantation.

## 4. Complement Activation

The complement system is activated during IRI through all three pathways (classic, alternative, and lectin). These pathways converge to the cleavage of C3 and C5, which leads to the formation of the membrane attack complex (MAC) and cell injury (Figure 1). More than a decade ago, a synthetic siRNA was developed to inhibit the messenger RNA of C5, resulting in the inhibition of complement activation and prevention of IRI [23]. However, clinical studies using eculizumab, a C5 blocking antibody approved for the treatment of paroxysmal nocturnal hemoglobinuria and atypical hemolytic uremic syndrome, challenge the clinical utility of C5. Kaabak et al. [24] demonstrated that a single dose one hour prior to graft reperfusion in pediatric kidney recipients had better early graft function and biopsy scores, but the incidence of graft rejection did not differ between groups. A later study in 2019 also showed that the administration of eculizumab prior to organ reperfusion showed no difference in delayed graft, function, and survival at 6 months [25]. Another critical component of the complement cascade is C3, which can be produced by many tissues. Danger signals released locally from apoptotic or ischemic cells can bind C3 through the alternative pathway. Using an experimental murine model, Farrar et al. [26] have demonstrated that local concentrations of C3 in proximal tubule epithelial cells are critical in mediating renal injury, rather than circulating C3 during renal ischemia. This is consistent with data that demonstrated that transplanting C3-deficient grafts results in indefinite allograft survival [27].

Human pre-clinical studies have demonstrated that the type of organ affects the role of complement activation and, in general, brain death augments its activation compared to living donation. Therefore, extended-criteria brain-dead donors and donation after circulatory death (DCD) are at higher risk of complement activation [28]. To prevent the detrimental effects of complement activation, Atkinson et al. [29] have used the complement inhibitor CR2-Crry in a mouse cardiac transplant model from brain-dead and living donors. The results confirmed an increase in complement deposition and inflammation in transplanted hearts from deceased donors compared to living ones. The inhibiting complement significantly reduced myocardial injury and prolonged graft survival. Recently, several randomized control trials have evaluated the therapeutic role of complement inhibition in preventing DGF associated with IRI. The complement inhibitor Mirococept was shown to reduce the local production of C3 in proximal tubule epithelial cells, diminish reperfusion damage, and significantly decrease DGF [30]. In addition, Jordan et al. [31] tested the effect of intraoperative administration of the C1 esterase inhibitor (C1-INH) berinert on DGF prevention in 105 recipients of DCD kidneys. Although the primary end point of DGF incidence was not met, significant reductions in need for dialysis and improvements in long-term allograft function were seen with C1-INH treatment. Recently, the long-term outcome of the trial has been updated and the researchers found that berinert treatment is associated with a lower incidence of graft failure [32]. These encouraging data suggest that complement inhibition is potentially valuable to prevent IRI damage in the clinic.

## 5. Danger Signals during Ischemia

The bridge between cellular death and innate immune activation is via intercellular communication of pro-inflammatory molecules. Cells release self-derived danger signals directly, such as Danger-Associated Molecular Patterns (DAMPs) and/or DNA into the extracellular milieu, which are recognized by Pathogen-Associated Molecular Pattern (PAMPs) expressed by innate immune cells (Figure 2).

DAMP signals are sensed by tissue macrophages and endothelial cells via Toll-like receptors (TLR), one of which is TLR4. Experimental IRI animal studies demonstrated that while the expression of TLR4 is associated with tissue damage and acts as a sentinel of acute damage [33,34], its deficiency in knockout mice is protective [35]. The activation of TLR4 leads to downstream signaling involving the release of NF-kB and its translocation to the nucleus, where it acts as a transcription factor to enhance the expression of inflammatory genes. Experiments with animal models have used siRNA, antisense oligomers, or drugs that directly inhibit NF-kB, which have been shown to curtail inflammation and prolong kidney graft survival [23,36,37,38]. Similarly, the therapeutic targeting of TLR4 with synthetic siRNA was shown to protect proximal tubular epithelial cells of the kidney from IRI using a mouse experimental model [39]. Although these approaches have not been evaluated in the context of organ transplantation, it is known that TRL4 is highly expressed in cadaveric donors, compared to living donors, and its activation contributes to chronic rejection [40]. Interestingly, changes in glucose and oxygen supply induced by ischemic injury that program myeloid cells metabolically and epigenetically to be hyper-responsive upon re-stimulation through TRL4 [41] and are associated with allograft rejection [42]. This innate response to secondary stimuli, also known as “trained immunity”, characterizes autoimmune disorders, and it is reasonable to suggest that it plays a role in IRI given that the underlying inflammatory mechanisms are shared [43,44,45].

Ischemia-primed endothelial cells following reperfusion are prone to mediate leukocyte adhesion and transmigration into tissues, which is another important function of the innate immunity in response to IRI. Endothelial cells express a set of molecules, such as P-selectin or intracellular adhesion molecule 1 (ICAM-1) to facilitate the transmigration of activated leukocytes into the interstitial space. Once activated, leukocytes release cytokines and proteases that promote tissue damage by increasing vascular permeability, thrombosis, and cell death [46]. After injury, several molecules participate in the interplay between leukocytes, endothelial cells, and platelets that facilitate immune activation. One of them is platelet-activating factor (PAF), which is a potent phospholipid mediator involved in acute inflammatory and immune responses. In murine kidney transplant models, it has been shown that PAF facilitates chronic allograft nephropathy and rejection [47]. During ischemia, endothelial cells also secrete substances such as platelet-derived growth factor (PDGF) that promote vasoconstriction to attenuate the tissue edema. At the time of reperfusion, this vasoconstriction is exacerbated by decreased endothelial nitric oxide synthase (eNOS) expression and increased sensitivity to vasoconstrictive molecules, including angiotensin II. As a result, there can be no reflow at the microcirculatory level following reperfusion, which has important clinical implications related to liver transplant. The liver parenchyma has a unique, dual blood oxygen supply (portal vein and hepatic artery). However, the bile ducts depend solely upon the hepatic artery and are very sensitive to ischemia. Given the current organ shortage and the need to expand the donor pool, there is an increased use of DCD liver grafts. These grafts have been shown to have more complications, predominantly ischemic-type biliary lesions, which are very difficult to treat and often require re-transplantation [4].

Another well-established inflammatory signal associated with of tissue injury is the C-reactive protein (CRP), which is an acute phase protein that circulates in plasma and whose levels rise in response to inflammation. CRP has been shown to play a critical role in leukocyte extravasation and tissue damage during renal IRI. Using immunohistochemistry and conformation-specific antibodies, Thiele et al. [48] were able to demonstrate localized conformational changes of CRP. Specifically, the study demonstrated that circulating pentameric CRP (pCRP) was shown to bind to activated biomembranes in the microcirculation of inflamed tissue. The pCRP-activated isoform is able to bind the leukocyte and facilitates its extravasation. Once in the tissues, pCRP is further altered and dissociates to monomeric forms (mCRP), which aggravate tissue injury by activating NADPH oxidase and generating ROS.

Further down the cascade of ischemic reperfusion damage, co-stimulation at the immunological synapse is of central importance. Following the two-signal model of T cell activation, the antigen is first presented by a cell on MHC, which binds to the T cell receptor; then, B7 binds to CD28 on the T cell surface. This allows T cell proliferation and cytokine production. In the absence of a second signal (no co-stimulation), the lymphocyte becomes anergic and undergoes apoptosis [49]. Recent studies in a mouse model of MHC class II mismatch cardiac transplant showed that B7:CD28 knockout mice had no change in graft survival. This is paradoxical, because the blockage of this pathway should lead to effective immunosuppression. However, it is thought that this same pathway is responsible for the maintenance of regulatory T cells, which are required for tolerogenicity [50].

B7 can also bind a receptor called CTLA4 on cytotoxic T lymphocytes. When this binding occurs, it overrides the two prior signals and prevents T cell activation [51]. Taking advantage of this phenomenon, a drug called belatacept was developed through the fusion of CTLA4 with the Fc constant region of human Ig. It can block the positive co-stimulation signal of B7-CD28, thereby inhibiting the immune response [52]. Two randomized clinical trials have used belatacept to target CTLA4. An earlier study showed it to be non-inferior to cyclosporine [53]. A second, larger study showed that post-transplant lymphoproliferative complications were more common, as well as a higher incidence of early acute rejection [54]. Studies approaching the efficacy of conversion from cyclosporine or tacrolimus to belatacept [55,56] or a regimen of belatacept with rabbit anti-thymocyte globulin [57] are ongoing. Thus, it remains unclear whether the use of belatacept is favored in routine clinical practice [58].

There are a number of inflammatory biomarkers reflecting innate immune system activation that can be detected in other fluids, such as urine, in acute kidney injury. Thus, KIM-1 (kidney injury molecule 1) or NGAL (neutrophil gelatinase-associated lipocalin) have been found to be increased in urine in mice and rat models after renal IRI and with different concentration changes during progression to acute or chronic kidney disease [59,60]. Curiously, NGAL secretion by hepatocytes occurs rapidly after lipopolysaccharide injection in mice, where it remains within the hepatic neutrophils, and it has been proposed as an IRI marker in liver transplantation [61]. On the other hand, NGAL may protect against acute liver injury and even promote liver regeneration by increasing hepatocyte secretion [62].

## 6. RNA Manipulation during IRI

For more than three decades, the use of synthetic antisense oligonucleotides that are taken up by cells in culture has been shown to inhibit the effects on cell DNA and protein synthesis through base-specific hybridization [63,64,65], which was named RNA interference (RNAi) [66]. In experimental models of IRI, this phenomenon has been used to target specific messenger RNAs (mRNA) and inhibit their translation. The use of small non-coding RNA molecules that silence RNA (microRNA) has been employed to regulate post-transcriptional gene expression. Consequently, microRNAs have emerged as key regulators of the immune response and may play an important role also in solid organ transplantation not only as therapeutic targets, but also as predictors of pathological states and injury markers [67,68] (Figure 1). Over the past decade, a growing number of miRNAs have been identified that we summarize in Table 1. With regard to IRI, Amrouche et al. [69] demonstrated that miR-146 knockout mice lacked protection from reperfusion at 14 days after insult in an experimental mouse model of IRI. In human kidney transplant recipients, miR-146 was shown to be differentially expressed in living and deceased donor allografts. Mechanistically, it was proposed that miR-146 is one of the genes transcribed by NF-kB during inflammation, but it acts as a negative regulator to repress IRAK1, which is the direct precursor to NF-kB. This ultimately leads to tubular injury and immune cell recruitment. In a recent study on human kidney transplant recipients, Khalid et al. [70] used an unbiased profiling approach to quantify post-transplant urinary microRNAs (Table 1). The results showed a predictive signature of DGF, which was validated in an independent cohort. Examples of these markers include miR-9, miR-10, and miR-21, the latter found in previous studies to be increased in tubular epithelial cells during ischemia to protect from apoptosis [71]. In the context of liver transplantation, several animal and human studies have identified microRNAs as biomarkers of hepatic injury and acute rejection [72,73,74]. Finally, recognize that extracellular RNA (eRNA) has been identified as a new type of alarmin that is released extracellularly during pathological conditions such as ischemia. By binding to TLRs, eRNA promotes the inflammation of immune and endothelial cells. It can be used as a diagnostic biomarker of ischemic damage during organ storage and machine perfusion or as a therapeutic target to be cleaved by RNase1 enzyme [75].

## 7. Organ Recovery and Processing

The period of storage and cold ischemia is an attractive platform for optimizing organ conditions prior to transplantation (Figure 1). In a retrospective review, prolonged cold ischemia (>36 h) was shown to be associated with decreased graft survival in renal transplantation, even if zero HLA mismatches were present. In other words, prolonged ischemia obviates the benefits of graft survival conferred by perfect histocompatibility match [76]. Consequently, there is a need for optimizing organ reconditioning to reduce early allograft injury, especially given that extended criteria for organ donation that includes DCD currently being used. To address this problem, extracorporeal organ perfusion has been implemented to reduce the metabolic stress during ischemia, which appears to reduce the incidence of biliary complications in long-term clinical trials. Sub-zero non-frozen preservation of liver was successfully developed in an experimental liver transplant in rats [77] and has been recently optimized for human studies with promising results [78,79]. Human livers were stored free of ice at −4 °C, extending the ex vivo life of the organ by 27 h with normothermic reperfusion with blood as a model for transplantation. A similar approach with a hypothermic oxygenated machine perfusion has been tried for liver transplantation under DCD conditions and is currently being evaluated in donation after brain death [80]. Interestingly, Eshmuninov et al. [81] recently developed an integrated ex vivo liver perfusion machine that integrates multiple core physiological functions, including an automated management of glucose levels and oxygenation, waste-product removal, and hematocrit control, which preserves functionality for up to 7 days. This crucial time window allows for the repair of injured livers, for the modification of immunogenicity, and removal of certain damaging metabolites described above. In the context of kidney transplantation, a prospective cohort study has identified a cluster of miRNA that is associated with ischemia reperfusion injury [82]. In pre-clinical animal models, more studies are taking place to evaluate temporal-specific gene changes and expression profiles after IRI that will create a databank to explore novel therapeutic approaches to prevent organ injury [83,84].

Preservation solutions are critical components of the extracorporeal organ perfusion, as they contain molecules aimed at providing metabolic supplies to mitigate organ damage related to ischemia. The University of Wisconsin (UW) solution is commonly used as hepatoprotective agent and has been shown to decrease IRI and improve short-term liver transplant outcomes [85]. The UW solution has been modified in several recent studies. Preoxygenated UW has been shown to be superior at sustaining ATP levels during cold ischemia static storage, which results in better long-term graft survival in a rat model of liver transplantation [86]. The addition of jun kinase (JNK) inhibitory peptides have been added to preservation solutions that inhibit stress-activated protein kinases, which reduce apoptosis in the context of pancreatic islet cell transplantation [87].

Machine perfusion has emerged not only as a way to diminish IRI and improve graft survival but also a way to administer specific drugs. This approach includes inhibition of pro-inflammatory molecules at the genetic level and blockage of receptors at the protein level. Several ones have been studied for off-label use during organ storage with no clear benefits as of yet. For example, etanercept, a TNFα inhibitor, has been administered ex vivo under machine perfusion hypothermia conditions in kidney transplant recipients, with no differences in DGF and graft survival between groups [88]. A recent study by Ritschl et al. [89] explored the effect of perioperative perfusion of extended-criteria kidney allografts with anti-T lymphocyte globulin (ATG), which is used routinely as induction therapy to prevent graft rejection, and the results demonstrated a reduction of DGF and the need for dialysis in the short term, but no therapeutic effect was observed at 1-year follow-up.

In addition to organ perfusion approaches to prevent IRI described above, the body has natural protective mechanisms against ischemia that can be leveraged to mitigate the effects of ischemia. A non-pharmacologic intervention that takes advantage of this fact is remote ischemic preconditioning (rIPC). In the clinical settings, rIPC entails for short repetitive periods of blood pressure cuff tightening, which acts as controlled limb hypoxemia that is protective of IRI (Figure 1). One of the molecular mechanisms for this appears to be the increase of RNase1, which degrades eRNA, therefore preventing TNFα and downstream immune activation [90]. In a large animal model of adult-to-pediatric kidney transplantation, rIPC was shown to decrease the rate of DGF and protect the glomerular filtration rate during the early postoperative period [91]. However, human studies have shown contradictory evidence. A randomized controlled trial with 225 kidney transplant recipients by Krogstrup et al. [92] showed no significant differences in DGF rates between rIPC groups and control groups. Presumably, this intervention would have more chances of success if applied to the donor allograft ex vivo instead of the recipient, but this is difficult to achieve in most surgeries, except for living donors, where the surgery is scheduled. A recent study, the REPAIR randomized controlled trial, evaluated rIPC in the context of living donor kidney transplantation and indicated promising results, as rIPC produced a sustained improvement in estimated glomerular filtration rate over the 5-year follow-up period [93]. Collectively, these studies indicate that reperfusion allows for the repair of injured organs during organ storage and machine perfusion, and they suggest that this period of time may be used to identify additional diagnostic biomarkers of ischemic damage that predict organ transplantation outcomes.

## 8. Cellular Regulation of Danger Signals

The bridge between the innate immunity and the adaptive immune response is mediated by myeloid cells. In the liver, resident macrophages (Kuppfer cells) and dendritic cells (DC) are activated by the inflammatory milieu and danger signals released during the initial injury. In the context of organ transplantation, activated macrophages and DC sustain the inflammatory cascade by secreting TNFα and IL-1β, but they also present foreign antigens to effector T cells to mediate transplant rejection (Figure 2). Interestingly, T cell activation can occur in the absence of antigen presentation, only with the pro-inflammatory milieu of innate the immunity during IRI [94]. Mechanistically, IRI upregulated the expression of CD40 in endothelial cells and macrophages, which bind to a ligand, CD40L (CD154), that is expressed on activated T cells. In animal and human studies, it has been shown that the CD40–CD40L receptor–ligand interaction mediates chronic graft rejection by increasing microvascular permeability [92] and the graft infiltration of inflammatory cells [95,96,97,98,99]. Experiments in rhesus monkeys successfully induced long-term graft acceptance despite major histocompatibility complex (MHC) mismatch by using an antibody against CD40L [100]; however, translational studies demonstrated that CD40L blockade induced thromboembolic complications. Thus, targeting CD40 receptor mRNA with RNA interference appears to be a better strategy [101].

Myeloid cells have also been shown to modulate T cell activity and to play a role in acute rejection. A randomized controlled study involving 109 kidney transplant recipients demonstrated that myeloid cells secrete danger signals, such as the calcium binding protein A9 (S100A) proteins that modulate the adaptive immune response toward tolerance [102]. A high expression of S100A9 in myeloid cells predicted better kidney transplantation outcomes in the 10-year follow-up of this study through mechanisms that involve the inhibition of DC maturation and T cell suppression. Therefore, it is possible that intracellular calcium increases during IRI discussed above, and it may cause myeloid cells to release S-100A8, causing them to shift from an immunogenic to a tolerogenic state [103]. Interestingly, myeloid cell secreting S100A9 express the myeloid-derived suppressor cells (MDSC) associated markers CD33, CD163, and DC-SIGN. MDSC have drawn particular interest in the field of transplant immunology, as systemic inflammation mobilizes immature myeloid cells from bone marrow into the circulation and the injured tissue where they may differentiate into MDSCs [104,105]. This is consistent with a recent study that reported the natural expansion of MDSCs following heart allograft transplantation in mice that favor graft survival prolongation [106]. In the context of renal IRI, a recent study by Yan et al. [107] evaluated the role of MDSCs using an experimental model and found that after induction by granulocyte colony-stimulating factor (G-CSF), MDSC secrete arginase-1 (ARG-1), which prevents T cell proliferation and potent immune responses. In this model, while MDSC would appear as protective, other studies have reported that MDSCs are mobilized by CRP to further exacerbate renal damage in IRI [108]. Therefore, it is not yet clear whether MDSCs play a predominantly tolerogenic role during IRI and peri-transplantation.

Hematopoietic Stem Cells (HSC) include a heterogeneous population of cells located in the bone marrow (BM) that differentiate in response to various stimuli and are critical for tissue homeostasis and repair [109,110,111]. The chemokine receptor CXCR4 and its ligand CXCL12 define an axis that regulates the BM trafficking of HSs. CXCL12, also called stromal cell–derived factor-1α (SDF-1α), is upregulated following inflammation [112,113]. In addition, several studies showed that the CXCL12/CXCR4 axis is upregulated during hypoxia and ischemic conditions [113]. Hu et al. demonstrated that the CXCL12/CXCR4 axis in myocytes and fibroblasts provides protection against myocardial IRI [114]. The mobilization of HSC is also promoted by G-CSF, and many tissues produce G-CSF in the presence of pro-inflammatory mediators such as TNFα, IFNβ, or IL-1β, which are all induced during surgery [115,116]. Tissue secreted G-CSF reaches bone marrow during IRI and induces the mobilization of HSCs that improve liver IRI recovery following transplantation by decreasing Kupffer cell activation [117,118]. While renal IRI injury can lead to a delayed recovery of transplanted kidney functions, these negative effects can be reversed by HSC through mitochondrial control, which decreases ROS production [118]. In addition, G-CSF provides a protective effect against IRI through different signaling pathways such as PI3K/Akt, STAT3, or endothelial NO synthase [119,120,121,122,123]. Other agents have been used in combination with G-CSF to enhance HSCs mobilization. For example, the granulocyte macrophage colony-stimulating factor (GM-CSF) has been shown to mobilize fewer cells than G-CSF [124], and AMD3100, a CXCR4 antagonist, has been used as adjuvant to favor G-CSF mediated mobilization of HSC in mice and humans [125,126,127,128]. Importantly, this method successfully mobilized enough cells in patients who previously failed mobilization with G-CSF [129]. In conclusion, HSCs have unique characteristics that facilitate organ transplantation recovery and attenuate the damage caused IRI, suggesting that HSC may change the cell composition of the graft and significantly impact organ transplantation [130].

## 9. Conclusions

Many of the current therapeutic approaches aimed at improving outcomes in organ transplantation focus on the pre-transplant and peri-transplant period, as the inevitable damage associated with the surgical procedure can result in poor long-term graft function. Insights into the molecular pathophysiology of IRI have opened the door to new therapeutic targets. Novel interventions such as succinate removal, ferroptosis inhibitors, regulation of complement cascade, and manipulation of regulatory cells, such as MDSC and HSC may play a role in reducing ischemia/reperfusion injury after transplantation. As long-term organ preservation technologies become more accessible, the possibility of organ reconditioning and manipulation with pharmaceuticals and novel strategies such as RNA interference is very attractive. In addition, as our understanding of how micro RNAs alter gene expression in ischemic tissue, their use as biomarkers seems promising. There changes associate the extent of injury and are predictive of future outcomes, which will also allow us to tailor treatment individually. Promising results from experimental and clinical studies are already underway (Table 2). As our knowledge of the mechanistic insights that govern IRI increases, the consequences of reducing IRI damage will not only be beneficial for individual patients but will also allow to reduce the number of patients on waiting lists, as previously suboptimal organs will be reconditioned. IRI develops throughout the different logistic stages of organ procurement initiated by hypoxia-induced metabolic stress and followed by multiple cellular death pathways and immune cell activation. Therefore, strategies that combine various treatments targeted at different mechanistic pathways during several time points will narrow the gap between the demand and supply of organs available for transplantation.

## Figures and Tables

**Figure 1 ijms-21-08549-f001:**
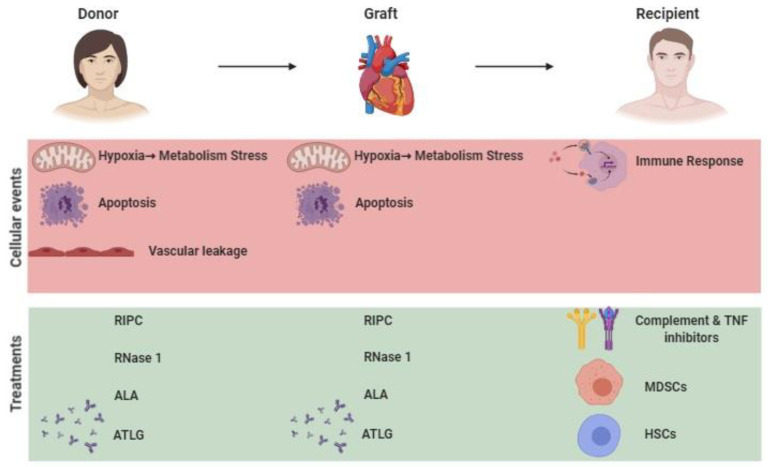
Cellular events involved in ischemia/reperfusion injury (IRI). (1) Hypoxia following death of the donor induces metabolic stress, vascular permeability, and cellular apoptosis. (2) In the ischemic graft, danger signaling pathways are potential therapeutic targets for optimization. (3) IRI generates a sterile inflammatory response in the recipient due to the events present in (2). Treatments based on the use of complement and (Tumor Necrosis Factor) TNF inhibitors are used to decrease the immune response, as well as cell therapy with myeloid-derived suppressor cells (MDSCs) and hematopoietic stem cells (HSCs), which are important in controlling this response.

**Figure 2 ijms-21-08549-f002:**
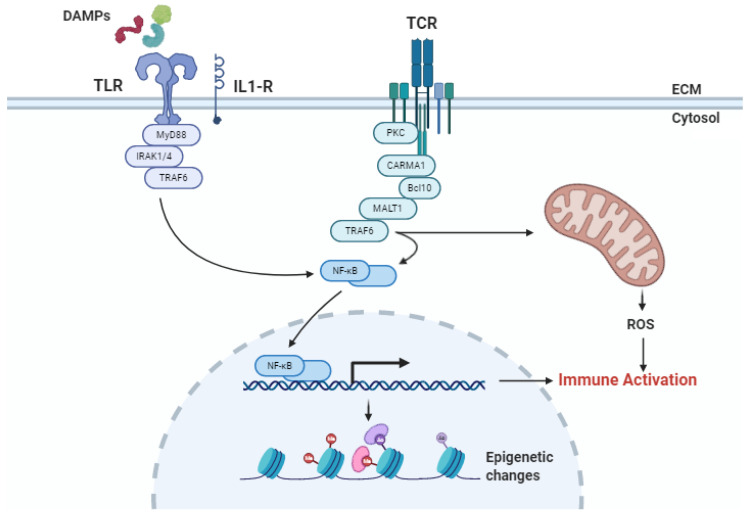
Signaling pathways involved in ischemia/reperfusion injury (IRI). (1) Damage-associated molecular patterns (DAMPs), released by injured and necrotic cells during IRI, are recognized by pattern recognition receptors (PRRs), such as Toll-like receptors (TLR), and Interleukin-1 receptor (IL1-R). The activation of PRRs results into the induction of nuclear factor-κB (NF-κB), which is a key regulator of DNA transcription, cytokine production, and pro-inflammatory signaling. As a result, DNA acquires epigenetic marks, and an immune response is initiated. (2) During IRI, T cells recognize specific antigens (Ags) through the T cell receptor (TCR), activating NF-κB and initiating an immune response without antigen-presenting cells. (3) The recognition of DAMPs or Ags leads to an increase of Nicotinamide Adenine Dinucleotide Phosphate Hydrogen (NADPH) oxidase activity in mitochondria, generating reactive oxygen species (ROS), which maintain the immune activation. Abbreviation: ECM, extracellular matrix; PKC, protein kinase C; CARMA, CARD motif of a subfamily of membrane-associated guanylate kinase (MAGUK) proteins called CARD-MAGUKs; MALT1, paracaspase MLT.

**Table 1 ijms-21-08549-t001:** Abbreviations. ITBL: ischemic type biliary lesions. DGF: delayed graft function. RT-PCR: reverse transcriptase polymerase chain reaction. AKI: acute kidney injury. IRI: ischemia reperfusion injury. dsRNA: double-stranded RNA.

Ref.	Author	Year	Model	miRNA	Effect	Detection Technique
[63]	Miller et al.	1979	synthetic	antisense oligonucleotides	Base specific hybridization	n/a
[64]	Nielsen et al.	1991	synthetic	antisense oligonucleotides	Base specific hybridization	n/a
[65]	Altmann et al.	1996	synthetic	antisense oligonucleotides	2nd generation	n/a
[66]	Fire et al.	1998	C. elegans	sequence specific post-transcriptional gene silencing	dsRNA interference	n/a
[71]	Godwin et al.	2010	Mouse	miR-21	Protective apoptosis kidney	Microarray + PCR
[72]	Farid et al.	2012	Human (*n* = 107)	miR-122, miR-148a, miR-192	Liver injury	RT-PCR (biased)
[74]	Hu et al.	2013	Rat	miR-192, miR-22	Liver injury	Microarray
[74]	Hu et al.	2013	Rat	miR-146	Acute rejection kidney	Microarray
[73]	Lankisch et al.	2014	Human (*n* = 88)	miR-517, miR-892a, miR-106a	ITBL	Microarray + PCR
[69]	Amrouche et al.	2017	Mouse, human	miR-146	AKI/IRI	RT-PCR (biased)
[70]	Khalid et al.	2018	Human	miR-9, miR-10, miR-21, miR-29a, miR-221, miR-429	DGF	Microarray

**Table 2 ijms-21-08549-t002:** DCD: Donation after Cardiac Death. TNF: Tumor Necrosis Factor. GVHD: Graft Versus Host Disease. HIF: Hypoxia-Inducible Factor. ECD: Extended Criteria Donor. RIPC: Remote Ischemic Preconditioning. ALA: Alpha Lipoic Acid. UC-MSC: Umbilical-Cord Mesenchymal Stem Cells. ATLG: Anti-T-Lymphocyte Globulin. Cr: Creatinine. ROS: Reactive Oxygen Species. eGFR: estimated Glomerular Filtration Rate. MAPC: Multipotent Adult Progenitor Cells. KTX: Kidney Transplant. LTX: Liver Transplant. CI: Cyclosporine. BPAR: Biopsy-Proven Acute Rejection. DGF: Delayed-Graft Function. DC: Dendritic Cell.

Reference	Year	Patients	Target	Mechanism in IRI	Intervention	Follow-up	Outcome	Clinical Notes
[53]	2005, Vincenti et al.	*n* = 218	CTLA4	Immune activation	Belatacept	1 year	BPAR 6mo	Non-inferior to CI
[54]	2010, Vincenti et al.	*n* = 686	CTLA4	Immune activation	Belatacept	1 year	Composite	Posttxp lymphoproliferative more common in Belatacept, higher early acute rejection
[102]	2016, Rekers et al.	*n* = 109 (KTX recipients)	Myeloid cells	Secrete S-100 and A6	Immunomodulation	10 years	Graft survival	S100 = less DC maturation = less T cell activity = better graft outcomes
[88]	2017, Diuwe et al.	*n* = 94	TNF alpha	Immune activation	Etanercept	3 years	Composite	Ex vivo, no differences between groups
[92]	2017, Krogstrup et al.	*n* = 225 (KTX recipients)	All pathways	Global	RIPC (BP cuff)	21 days	Time to 50% drop in plasma Cr	No sig. differences, RIPC protocol not optimized?
[85]	2017, Aliakbarian et al.	*n* = 115 (LTX recipients)	*N*-acetylcysteine	Hepatoprotective	Adding to UW solution	Hospital stay	Postreperfusion hypotension	Hypotension after reperfusion more common in experimental group
[31]	2018, Jordan et al.	*n* = 105 (KTX recipients)	C1 esterase inhibitor	Vascular leakage	Intra-op C1 inhibitor	1 year	Graft function	DGF is IRI-induced
[24]	2018, Kaabak et al.	*n* = 57 (KTX recipients)	C5b-9	Immune activation	Pre-op Eculizumab	3 years	Graft function	Better early graft function and biopsy scores but unacceptably high number of early graft losses
[10]	2018, Casciato et al.	*n* = 40 (LTX recipients)	Alpha lipoic acid	Antioxidant (HIF-alpha)	ALA before reperfusion	30 days	Gene changes to hypoxia/ROS	qPCR for panel of postreperfusion genes, alarmins
[130]	2018, Sun et al.	*n* = 42 (KTX recipients)	Allogenic MSC	Immunomodulation	UC-MSC before and during Txp	1 year	Graft function	Novel cell-based approach, delivery not optimized yet (renal artery?)
[89]	2018, Ritschl et al.	*n* = 50 (KTX recipients)	Periop organ	ECD, DCD grafts	Perioperative perfusion with ATLG	1 year	Need for dialysis	No change in long term
[93]	2019, Veighey et al.	*n* = 406 KTX live donor pairs	Global	Global	RIPC = cuff to limb	5 years	Early eGFR, 5yr-survival	Living donation is a scheduled surgery, much easier to arrange for RIPC.

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
