# Peer review of "Review: Ischemia Reperfusion Injury—A Translational Perspective in Organ Transplantation"

_ijms, 2020, doi:10.3390/ijms21228549_

Round 1

Reviewer 1 Report

The background, pathways of IR injury and the updated literature on mitigating IR injury in organ transplantation is well presented.

Unfortunately the more recent machine perfusion studies are still at early stages and we don't have robust randomized trials. Overall interesting review 

Author Response

The background, pathways of IR injury and the updated literature on mitigating IR injury in organ transplantation is well presented. Unfortunately, the more recent machine perfusion studies are still at early stages and we don't have robust randomized trials. Overall interesting review.

We appreciate the reviewer 1 comments. We recognize that organ shortage is a critical problem and that by conditioning and utilizing suboptimal organs affected by IRI in specialized perfusion machines is essential to increase the pool or organs available for transplantation. Indeed, while it is a promising approach, machine perfusion for organ reconditioning, biomarkers and point-of-care diagnosis is still at early stages. We have further addressed this issue on point “7. Organ recovery and processing” on pages 12-13.

Reviewer 2 Report

The authors summarized the recent topics in ischemia-reperfusion injury focusing on organ transplantation. However, the story of this work is relatively difficult to read throughout the entire review. Because the topics included in this manuscript are wide, the story would be revised in more straight forward topics.

In addition, the present items are collected as mechanistic series, however, sentences relating liver, pancreatic and kidney transplantation are included in each item. This reviewer fells preferable to see the topics each organ, because the damage and molecular mechanisms against ischemia-reperfusion injury may be different among organs. For example, some urinary biomarkers had been found in the ischemia-reperfusion injury model mice. The expressions of KIM-1, NGAL and L-FABP are elevated in that treatment, and each molecule contribute the recovery of damaged kidney. However, in the present manuscript, any urinary biomarker is cited.

The references for BMT are relatively old era compared to the organ transplantation. Because the title of the manuscript includes “organ transplantation”, and therefore, the topics about BMT can be deleted in the present work.

Others:

The authors pay effort to make more attractive abstract covering these topics for examples.

Page 16, line 341 The phrase “minimizing IRI” is not specific.

Author Response

The authors summarized the recent topics in ischemia-reperfusion injury focusing on organ transplantation. However, the story of this work is relatively difficult to read throughout the entire review. Because the topics included in this manuscript are wide, the story would be revised in more straight forward topics.

We agree with the reviewer’s comment and we have modified the text throughout the manuscript to ensure that readers can easily follow the rationale of the review article. Article sections have been modified to discussing ischemia-reperfusion pathogenic mechanism, according to the scope of the International Journal of Molecular Sciences journal.

In addition, the present items are collected as mechanistic series, however, sentences relating liver, pancreatic and kidney transplantation are included in each item. This reviewer fells preferable to see the topics each organ, because the damage and molecular mechanisms against ischemia-reperfusion injury may be different among organs. For example, some urinary biomarkers had been found in the ischemia-reperfusion injury model mice. The expressions of KIM-1, NGAL and L-FABP are elevated in that treatment, and each molecule contribute the recovery of damaged kidney. However, in the present manuscript, any urinary biomarker is cited.

We appreciate this critique. However, the authors felt that, while topics could have been arranged by organs transplanted, there would have been repeating mechanisms in each topic, as the mechanisms underlined in the review and are shared by most organs. We have included the suggested urinary biomarkers found in the mouse model of IRI (pages 9-10, lines 189-194).

The references for BMT are relatively old era compared to the organ transplantation. Because the title of the manuscript includes “organ transplantation”, and therefore, the topics about BMT can be deleted in the present work.

We agree with the reviewer and references related to BMT have been deleted from the manuscript.

 Page 16, line 341 The phrase “minimizing IRI” is not specific.

The text has been modified accordingly.

Round 2

Reviewer 2 Report

The authors have been revised the manuscript. Most of my criticisms were treated adequately. However, the revised abstract seems NOT suitable for the manuscript. It seems an introduction for the introduction of main text.

Then, the authors should revise abstract more attractively citng the topic in the text.

Author Response

Dear Reviewer, 

thank you very much for allowing us to review the manuscript. We agree that the abstract was not focused to the aim of the review. We have changes it accordingly.

You will find following the abstract. The changes are indicated in the word file in blue colour background in the abstract. We include all the manuscript with the changes.

"Thanks to the development of new, more potent and selective immunosuppressive drugs together with advances in surgical techniques, organ transplantation has emerged from an experimental surgery over fifty years ago to being the treatment of choice for many end-stage organ diseases, with over 139,000 organ transplants performed worldwide in 2019. Inherent to the transplantation procedure is the fact that the donor organ is subjected to blood flow cessation and ischemia during harvesting, followed by preservation and reperfusion of the organ once transplanted into the recipient. Consequently, ischemia/reperfusion induces a significant injury to the graft with activation of the immune response in the recipient and deleterious effect on the graft. The purpose of this review is to discuss and shed new light in the pathways involved in ischemia reperfusion injury (IRI) that act at different stages during the donation process, surgery and immediate post-transplant period. Here, we present strategies that combine various treatments targeted at different mechanistic pathways during several time points to prevent graft loss secondary to the inflammation caused by IRI."

Thank you very much for your help regarding this issue.

Looking forward to hearing from you.
